# Motivations of Households towards Conserving Water and Using Purified Water in Czechia

Roman Lyach * and Jiří Remr

Department of Environment, INESAN (Institute for Evaluations and Social Analyses), Sokolovská 351/25, 186 00 Prague, Czech Republic
* Correspondence: roman.lyach@inesan.eu

**Abstract:** The need to assess reasons why households do not want to conserve water or use purified water is critical when facing water scarcity during the climate change crisis. This study aimed to provide an analysis of perceptions of the public in Czechia towards water conservation. A representative questionnaire survey ($n$ = 1824) was conducted in the whole Czech Republic to see why households hesitate to conserve water and use purified water. We discovered that most household owners are interested in conserving water and some of them are interested in using purified water. The household owners are willing to conserve water if it does not cost them too much time or comfort. They are mostly willing to think about using purified water for purposes that are hygienically safe. They mostly trust people and public figures that are closest to them, and they consume media like television and the Internet. We recommend that any communication campaign aiming to persuade households to conserve water and use purified water should explain how to effectively conserve water in households. It should also explain that using purified water in households is not unhygienic and is almost completely safe.

**Keywords:** public perceptions; public behaviour; environmental psychology; Czechia; water conservation; purified water; reused water





## 1. Introduction

Droughts have become one of the most important environmental issues of the 21st century. Water conservation and usage of purified water are among the most efficient ways for the mankind to adapt to droughts that are connected to climate change [1]. However, if we want to be successful at fighting droughts, the public needs to be on board with conserving water and using water purification technologies [1]. Therefore, it is critical to assess the reasons why households are not willing to conserve water and use water purification technologies. In households, conserving water means using water-saving technologies and practices during washing and cleaning, while water purification means using water from sink and shower to flush the toilet [1–6].

There are many studies that have analysed the perceptions of households towards water conservation and using purified water in many states around the globe. They have mostly found that people are willing to conserve water and use purified water, especially in countries that are geographically located in regions with water scarcity [2–6]. Household representatives are mostly willing to conserve water if it does not cost them too much money or time. They are mostly willing to use water-conserving technologies (dishwashers, showers, washing machines). They also like to use rainwater to water plants and gardens. Conversely, they are not willing to save water on activities that are too costly or time consuming. The motivations for conserving water are both practical and environmental [2–6].

Conversely, the acceptance of water purification technologies is lower due to limited know-how regarding the technology among households. Household representatives are willing to use purified water for actions that are hygienically safe (cleaning houses, washing

cars, watering gardens). However, they are not willing to use it for hygiene-related actions like drinking, washing dishes, showering, cleaning hands, etc. People mostly fear purified water because they think it has residual pathogens, smell, colour, and other unhygienic components [7–16]. Nevertheless, households in countries with extreme water scarcity (e.g., Africa or Middle East) have high acceptance for using purified water due to the necessity to conserve as much water as possible [17–19]. This is because they do not have a choice, as their fear of not having water at all trumps their fear of residual pathogens.

In addition to analysing perceptions of households towards water conservation, it is important to create effective PR (public relations) and communication strategies that target specific groups of people who have not yet decided if they want to act pro-environmentally but can be persuaded to do so. There are already several studies that have identified effective PR strategies towards water conservation and usage of purified water. They found that if we want to persuade households, we need to identify their representatives based on socio-economic factors, their perceptions towards water conservation, and their values [20–27].

While in general, the perceptions of households towards conserving water and using purified water are well-known, there are no data from Czechia on this topic. The global scientific literature is lacking an up-to-date sociological study that would describe perceptions of households towards water conservation in Czechia. If we want to understand why households want to conserve water, we need to perform a sociological study to analyse how many households want to conserve water. In addition, we need to link it to a psychological study to analyse why do other households not want to conserve water. Moreover, the scientific literature is also lacking the basis for a good communication strategy towards the households that could persuade them to conserve water.

*Aims of the Study*

This study aimed to fill the knowledge gap by providing a representative study that analyses the perceptions of household representatives in Czechia (the target group) towards water conservation and usage of purified water in households. This study then further links the perceptions of the household representatives to socio-demographics, level of trust, life values, and media consumption of the household representatives. We wanted to know why household representatives hesitate to conserve water and use purified water. We hypothesised that the reasons are mostly based around hygiene, know-how of technologies, and practical reasons. We also hypothesised that water-conserving households will differ from the public in socio-demographics, trust, life values, and media consumed.

## 2. Materials and Methods

### 2.1. Study Area and Data Collection

The study was carried in Czechia (51°03′20″–48°33′06″, 12°05′26″–18°51′33″) in central Europe (Figure 1). Czechia is located on the main European watershed, and all water is drained into one of the three seas (Baltic Sea, Black Sea, or North Sea). High quality drinking water is available to all inhabitants without restrictions, although there are minor water shortages during the summer in small villages in the countryside. There are no large natural lakes located in Czechia, and the majority of drinking water is taken from rivers.

The survey research was conducted in Czechia during April and May 2021 using questionnaire surveys. The target population is the general population aged 18–79, comprising residents permanently living in Czechia.

To assure the representativeness of the sample, the rigorous multistage random procedure using address-based sampling was applied. Since no adequate sampling frame (e.g., the citizens´ register or list of residents) is available in Czechia, the primary sampling units were selected. Within each primary sampling unit, addresses were identified, and an appropriate number of households were selected. Finally, the interviewers visited the pre-selected addresses and attempted to identify the prospective respondents using the Kish-table [28].

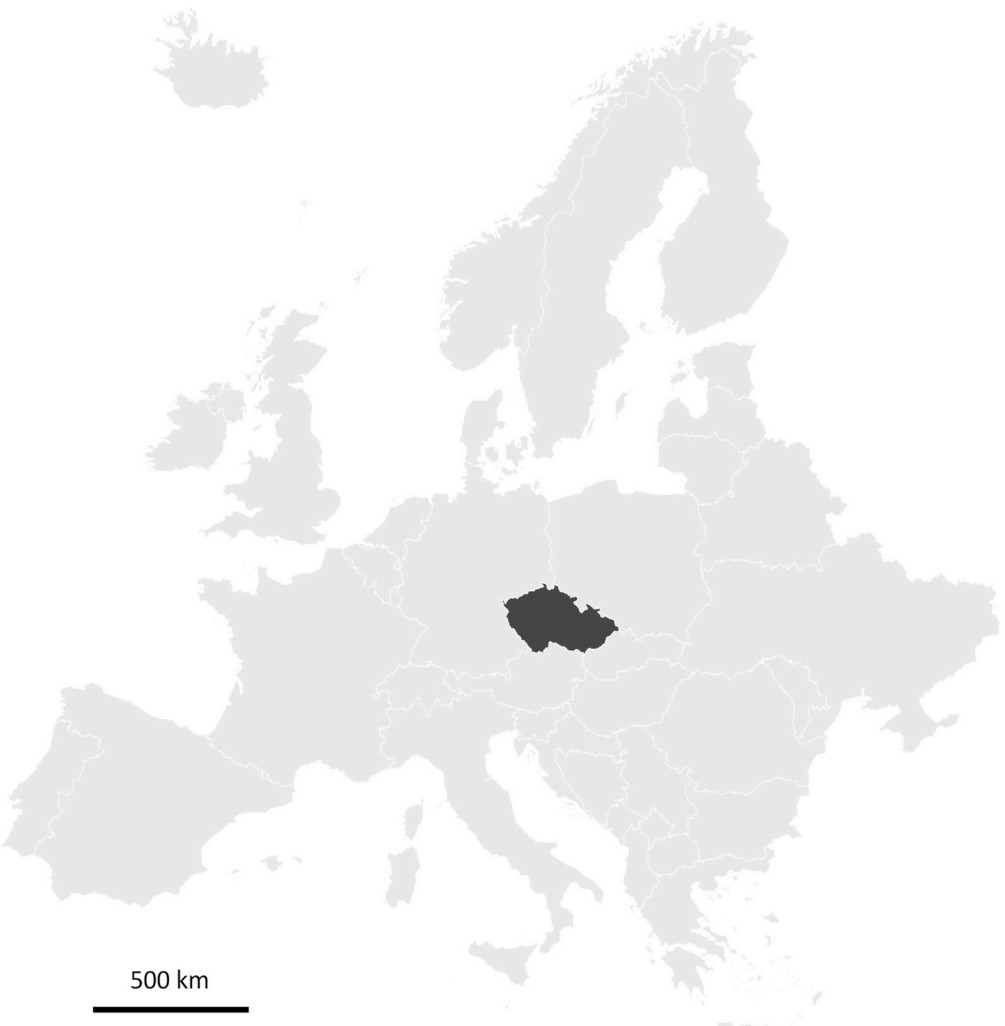

**Figure 1.** Map of the study area where the study was conducted in April–May 2021 (Czechia, 51°03′20″–48°33′06″, 12°05′26″–18°51′33″) in central Europe.

Altogether, 243 primary sampling units throughout Czechia were selected; within each of these units a maximum of 15 addresses were identified. Interviewers contacted 3413 households and performed 1843 interviews; thus, the AAPOR-5 response rate is 53.9 percent. However, due to incompleteness of some of those interviews, when respondents refused to provide key socio-demographic data, the datafile comprised 1824 cases that was used for the analyses.

Data collection took the form of standardized face-to-face interviews (or PAPI—pen and paper interviewing) when well-trained interviewers asked the questions read from the paper questionnaire and recorded the respondents´ answers. The average time spent to complete the questionnaire was 70 min. Table 1 shows the structure of the sample with respect to gender, highest achieved education, occupation, household net monthly income, socio-economic status, size of settlement, and type of residence.

From all 1824 cases used for the analyses, 40 percent were supervised by check-backs and verified in terms of compliance with ethical and quality standards. Moreover, all questionnaires (100%) were double-punched and checked for possible transcription errors, and the compliance with the instructions (screening procedure, skipping patterns, etc.) and the response errors resulting in incongruence of socio-economic data were also checked.

**Table 1.** Socio-demographic distribution of respondents who were chosen to be representative of the Czech socio-demographic distribution. The data were collected in Czechia in April–May 2021 using questionnaire surveys (1824 respondents).

| Gender | % |
|---|---|
| male | 50 |
| female | 50 |
| **Education** | **%** |
| elementary | 7 |
| high school without maturity exam | 32 |
| high school with maturity exam | 46 |
| university | 15 |
| **Occupation** | **%** |
| employee | 62 |
| self-employed | 9 |
| student | 5 |
| economically inactive | 4 |
| senior | 20 |
| **Monthly household income** | **%** |
| under EUR 400 | 1 |
| EUR 400–600 | 4 |
| EUR 600–700 | 4 |
| EUR 700–800 | 5 |
| EUR 800–1000 | 7 |
| EUR 1000–1200 | 8 |
| EUR 1200–1400 | 18 |
| EUR 1400–2000 | 30 |
| EUR 2000–2500 | 17 |
| over EUR 2500 | 7 |
| **Socio-economic status** | **%** |
| socially weak class | 12 |
| lower class | 14 |
| lower middle class | 39 |
| upper middle class | 25 |
| high class | 11 |
| **Size of the city of residence** | **%** |
| under 10,000 inhabitants | 49 |
| 10,000–20,000 inhabitants | 9 |
| 20,000–50,000 inhabitants | 12 |
| 50,000–100,000 inhabitants | 8 |
| over 100,000 inhabitants | 22 |
| **Type of residence** | **%** |
| owned house or flat | 71 |
| rental house or flat | 29 |

*2.2. Data Collection Tool (Questionnaire)*

The questionnaire was based on excessive literature reviews in the field of public perceptions of water conservation and the use of purified water [1–19,29–31]. All questions covered the following major areas:

(1) Socio-economic and socio-demographic descriptors

Overall, 15 items were used to identify the demographic, social, and economic background of each respondent. The questions focussed, among others, on age, highest achieved education, occupation, household net monthly income, and living conditions of the respondents. These variables were consequently used as the independent variables in differentiat-

ing the relevant subpopulation, and we hypothesised these characteristics explain some attitudes and behavioural patterns of the respondents.

(2) Motivations to conserve water and use purified water in households

Altogether, 68 items were asked to measure the intentions and declared behaviour. The specific questions were focused on the willingness to perform specific activities to help to conserve water (e.g., hand washing, house cleaning, lawn watering, car washing, showering), on the reasons to conserve water (local and global scarcity of water), on the willingness to use purified water for specific purposes (e.g., washing, cleaning, lawn watering), and on perceptual barriers of purified water usage (especially smell, colour, pathogens, hazards, price, know-how).

(3) Trust in key stakeholders (opinion leaders) that might persuade the others to change the attitudes and behaviour with respects to conserving water and using purified water in households

The credibility of 14 different opinion leaders were measured, including the public bodies (for instance department of state, municipal office, and university), enterprises (e.g., local utility company), and individual persons (neighbours, relatives, friends, etc.).

(4) Media consumption

Detailed questions were asked to determine what media the respondents follow and how frequently.

(5) Social and life values

In total, 182 items were used to measure the social and life values of the respondents. Such data provide an overall attitudinal, perceptual, and emotional background that helps to explain the key drivers of individual behaviour. Among others, the System Justification Scale or Eco-attitude Scale was used.

### 2.3. Data Analysis

The statistical software SPSS was used for the data analysis. We performed exploratory data analysis and produced descriptive statistics for each variable; values of nominal and ordinal variables were compared among the key subgroups and tested with the chi-square test. Ratio variables were first checked for normality distribution and the means were then compared among the relevant subpopulations; we also used ANOVA (post-hoc). The reliability of scales was tested with Cronbach's alpha coefficient; complex constructs reflecting the attitudes of respondents were analysed with exploratory factor analysis. We used exploratory factor analysis (EFA) with varimax rotation to identify latent (hidden) factors and the main reasons why household representatives conserve or do not conserve water and use or do not use purified water. We considered any eigenvalue variable loading greater than 0.40 as important. The scores for each of the factors were calculating by summing the z-scores of the individual items for each factor. All statistical tests were conducted at a 0.05 level of significance.

### 3. Results

*Perceptions and Behavioural Patterns of the Target Group*

We found that the majority of the sample was already conserving water in their households. Only a small part of the respondents did not care about saving water, while about 20% of the interviewed persons were undecided (Figure 2A). The willingness to use purified water in households was a bit lower: only 17% of homeowners were thinking about using purified water. The majority of the sample was willing to save water but not to use purified water. About 20% of the interviewed homeowners cared neither about saving water nor about using purified water (Figure 2B).

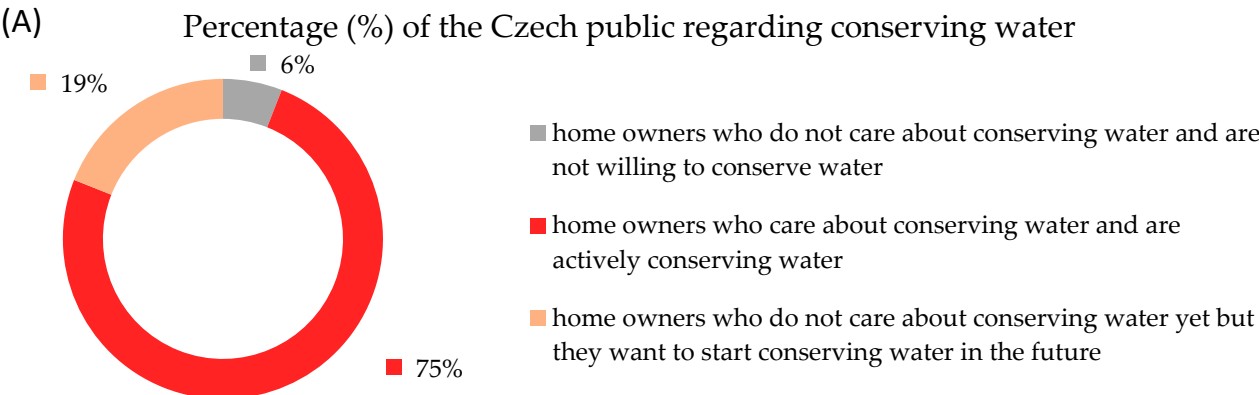

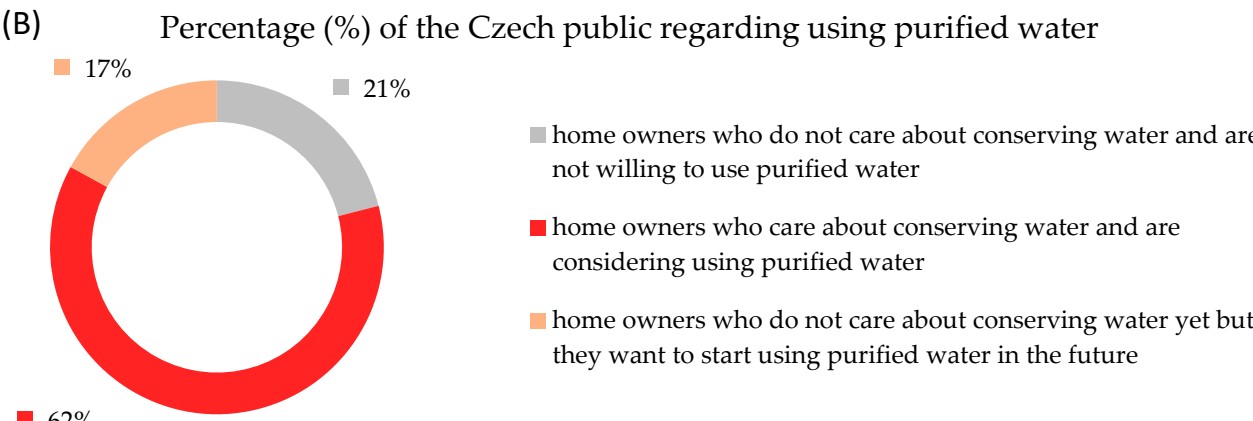

**Figure 2.** Distribution of the public regarding their willingness to conserve water in households (**A**) and use purified water in households (**B**). The data were collected in Czechia in April–May 2021 using questionnaire surveys (1824 respondents).

The study showed that respondents who are willing to conserve water and use purified water make up a heterogeneous group. Their socio-economic characteristics are mostly like those of the rest of the public (Figure 3). The groups of people who are interested in conserving water and using purified water do not significantly differ from the public in gender, age, socio-economic status, occupation (type of job), or type of living ($p > 0.05$ for each factor). However, they can be distinguished from the public by being more pro-environment and by accepting their personal role in nature conservation.

Our target group of household owners was mostly willing to conserve water using techniques that are not harmful to their health or well-being. For example, activities like washing dishes or clothes using water-saving programs had high acceptance, and so did showering or using rainwater for watering plants. Conversely, activities that require a lot of work or are potentially unhygienic had lower acceptance (Figure 4). The only reason why homeowners are not willing to save water is that they think that there is enough water in their local city or village and so do not feel the urge to conserve water. The most important factor that influenced the willingness to conserve water was practical usefulness and comfort (0.828), followed by financial profit (0.852), and hygiene (0.523).

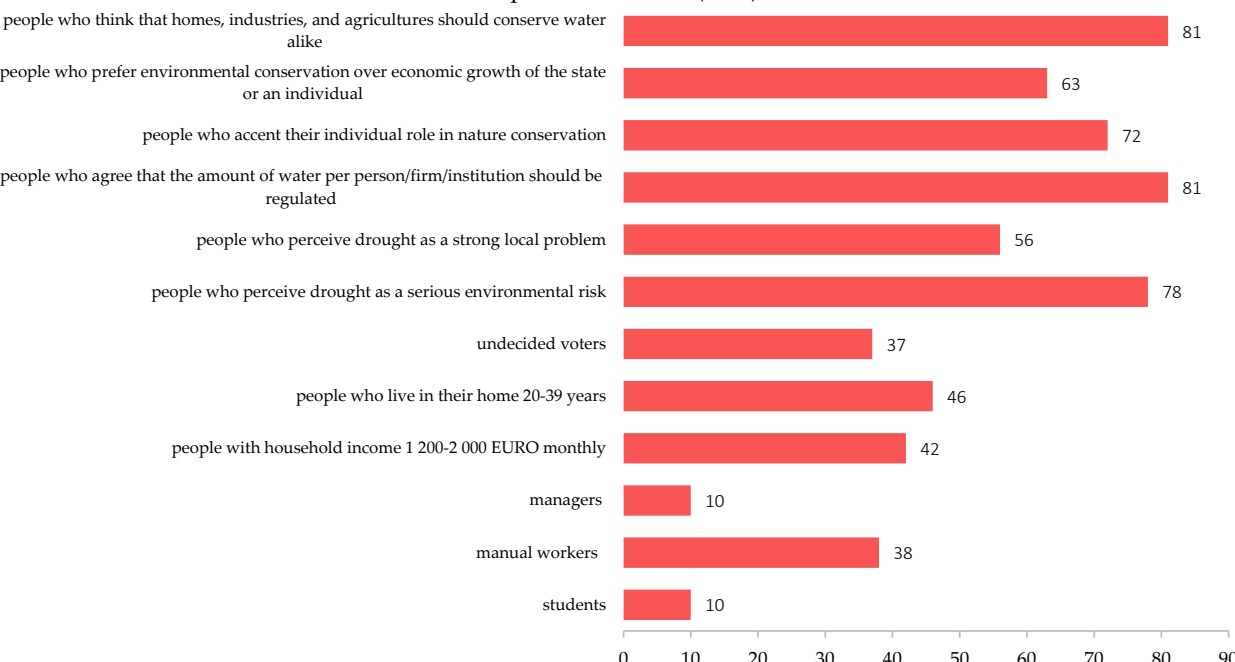

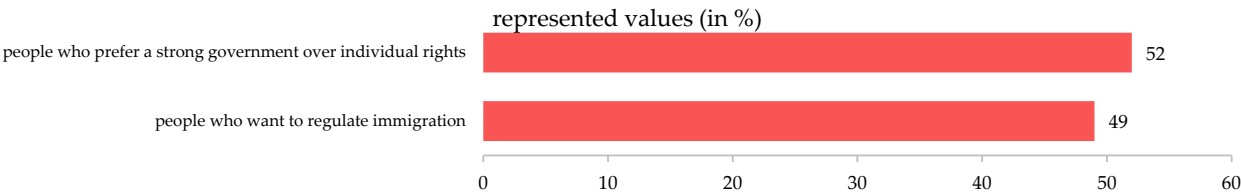

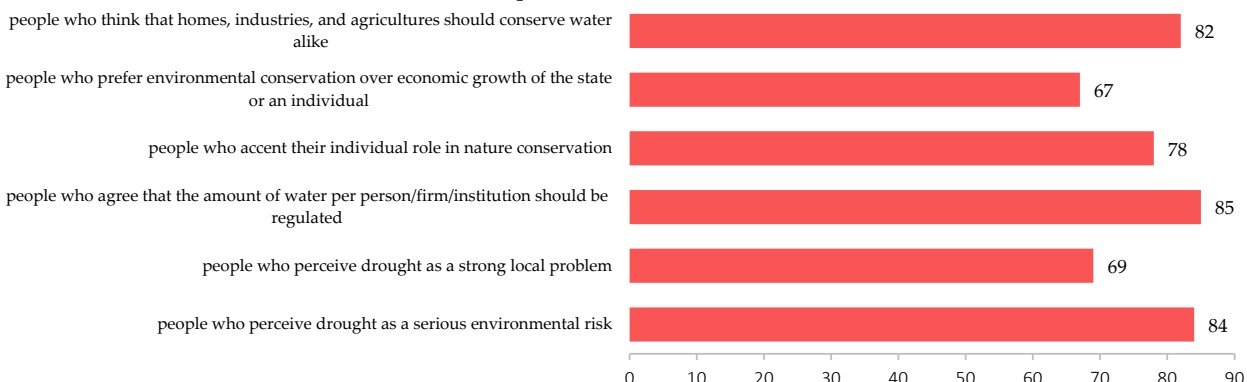

**Figure 3.** *Cont.*

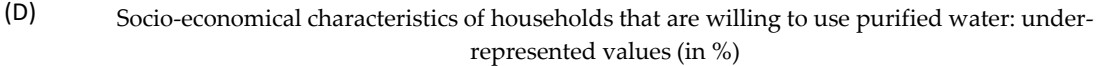

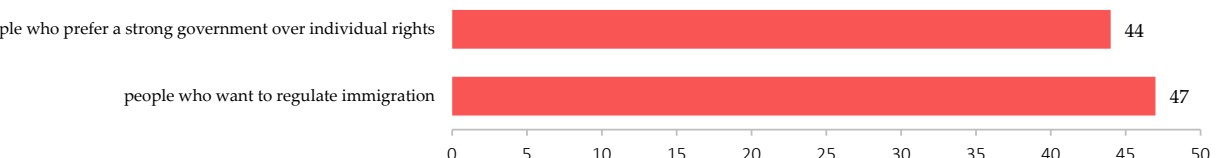

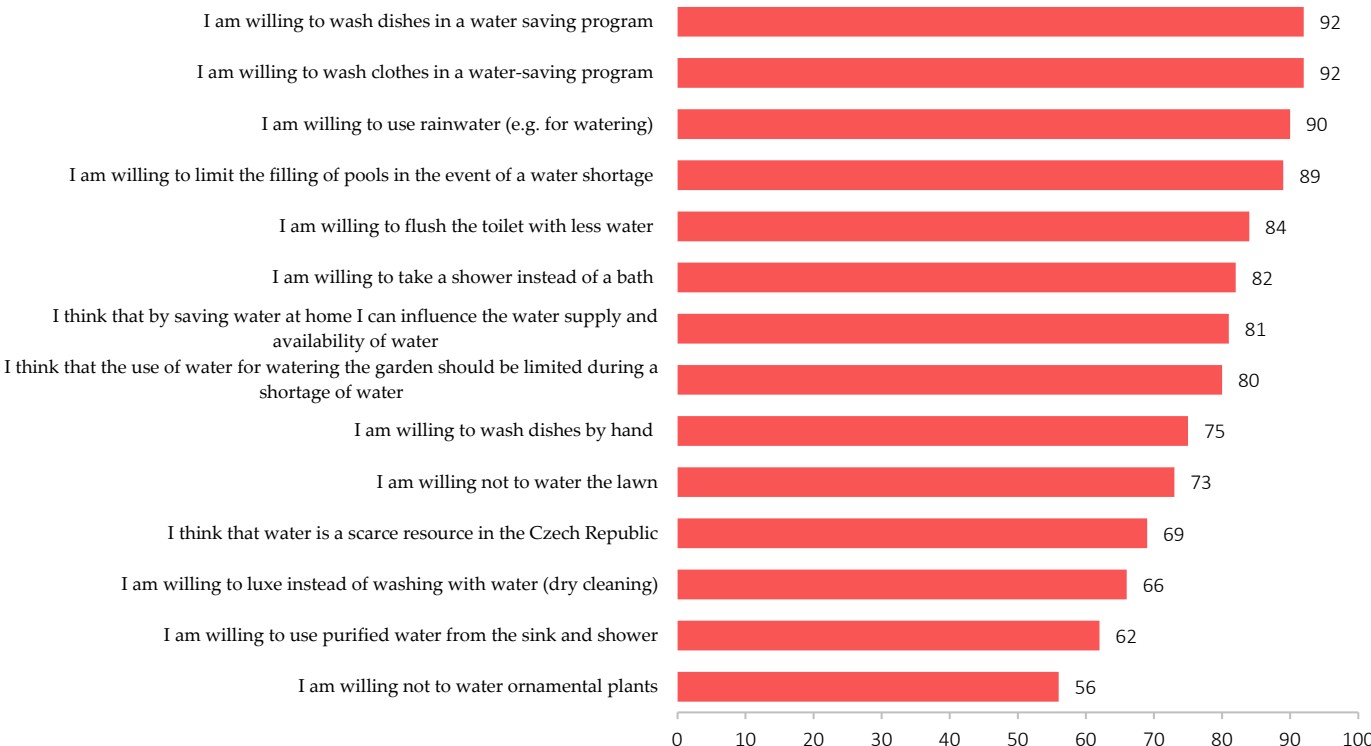

**Figure 3.** Socio-economic characteristics that separate water-conserving homeowners from the rest of the public. The graphs (**A,B**) show which socio-economical characteristics are significantly more and less represented, respectively, in water-conserving household owners in comparison to the public. The graphs (**C,D**) show which socio-economical characteristics are significantly more and less represented, respectively, in households willing to use purified water in comparison to the public. The number behind each column represents the percentage of households that fit the description. The data were collected in Czechia in April–May 2021 using questionnaire surveys (1824 respondents).

**Figure 4.** The main activities that household owners claim that they are willing to do to conserve water. The number behind each column represents how many respondents agreed with the statement (in %). The data were collected in Czechia in April–May 2021 using questionnaire surveys (1824 respondents).

Similarly, the homeowners are mostly for using purified water, but they are only willing to use it for activities that are hygienically safe. They are very much willing to use purified water for flushing toilets, washing cars, cleaning rooms, and watering lawns. Conversely, they are not willing to use purified water for drinking, cooking, hygiene, or other activities that involve getting into contact with the water (Figure 5). They are also concerned about potential hygienic hazards of the purified water, e.g., residual smell, bacteria, and impurities (Figure 6). The most important factor that influenced the willingness to use purified water was hygiene (0.903), followed by comfort and practical usefulness (0.872). The most important factor that described why people do not want to use purified water was hygiene (0.854) followed by practical usefulness (0.816).

The willingness of people to use purified water on specific actions (in %)

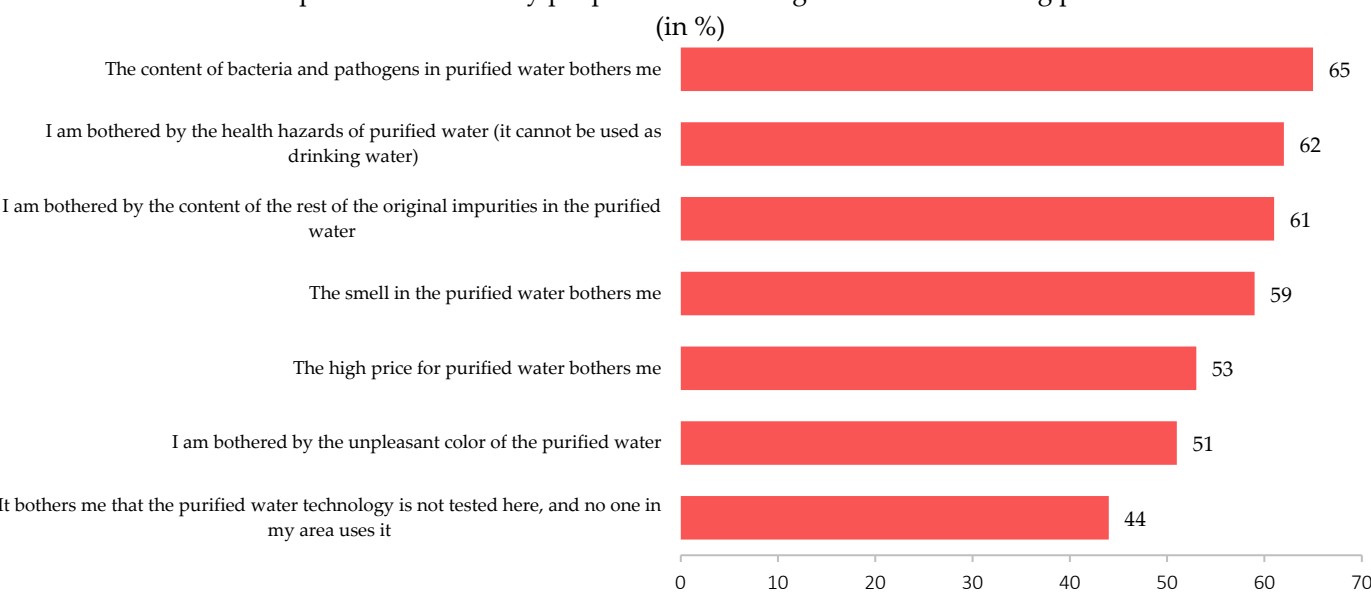

**Figure 5.** The main activities that household owners claim that they are willing to use purified water for. The number behind each column represents how many respondents agreed with the statement (in %). The data were collected in Czechia in April–May 2021 using questionnaire surveys (1824 respondents).

The most important reasons why people are not willing to think about using purified water (in %)

**Figure 6.** The main reasons why household owners claim to refuse to use purified water in their households. The number behind each column represents how many respondents agreed with the statement (in %). The data were collected in Czechia in April–May 2021 using questionnaire surveys (1824 respondents).

The homeowners from our study group mostly trust the opinions of the people closest to them (Figure 7). Family and relatives were the people they trust the most. From political figures, homeowners frequently trust the local major and local authorities the most. They also trust scientists and water experts. Conversely, celebrities and the European commission organisations have little trust from these households. The most important factor that influenced trust was personal closeness (0.883) followed by recognized expertise (0.746).

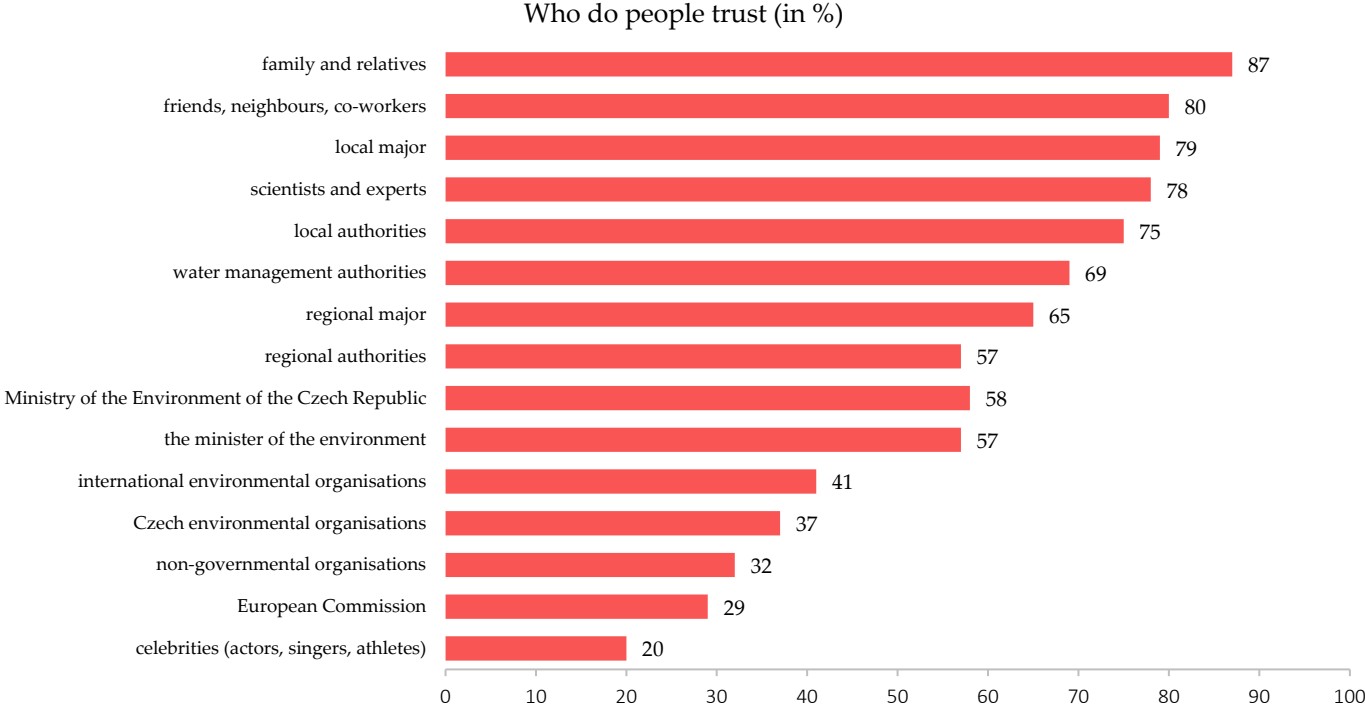

**Figure 7.** Level of trust that water-conserving household owners give to individual people and institutions. The number behind each column represents how many respondents trusted the subject (in %). The data were collected in Czechia in April–May 2021 using questionnaire surveys (1824 respondents).

As far as the media are concerned, household representatives mostly watch and trust the Internet news but also state TV, social media, and local newspapers (Figure 8).

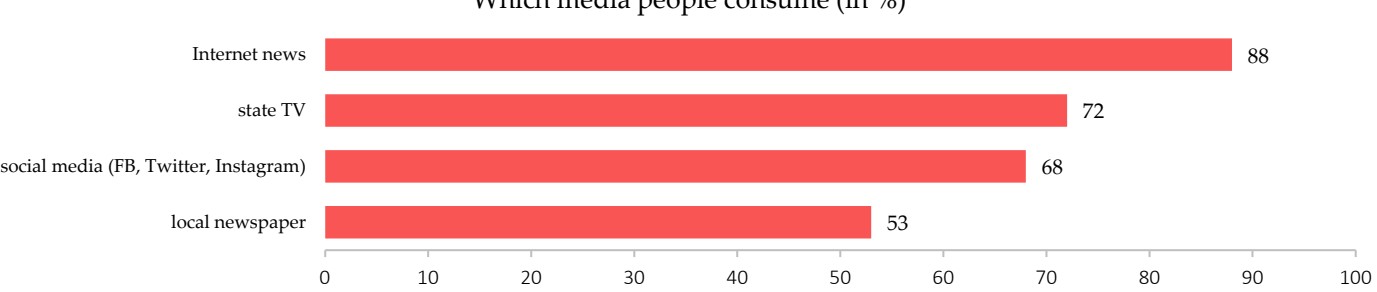

**Figure 8.** Which individual media the household representatives regularly watch and consume. The number behind each column represents how many respondents follow the media (in %). The data were collected in Czechia in April–May 2021 using questionnaire surveys (1824 respondents).

## 4. Discussion

We discovered that most household owners are interested in conserving water and some of them are interested in using purified water. The household owners are willing to conserve water if it does not cost them too much time or comfort. They are mostly willing to think about using purified water for purposes that are hygienically safe. They mostly trust people and public figures that are closest to them, and they consume media like television and the Internet.

The main discovery that household owners are willing to conserve water and use purified water if it is safe, easy, and hygienic is well in line with the results of other sociological and psychological studies [1–19]. Only 6% of households are unwilling to conserve water, which is a surprisingly low number in comparison to the numbers in similar sociological studies [1–19]. This is perhaps because drought has become an important topic in the media, and people are starting to realise the seriousness of droughts. It seems that there is no need to persuade people to conserve water since they are already on board with it.

Conserving water is generally well perceived among the public, not only in Czechia but also globally. Using purified water in households is, however, a different story. Czech household owners only marginally support it, mostly because water is not scarce in central Europe. Over 20% of people do not trust purified water, which accounts for 2 million people in Czechia. There is a need to explain to those people that purified water is safe, otherwise, they will not accept its usage.

This is different from other countries like Israel that have water scarcity issues and therefore higher acceptance for using purified water [17]. While using purified water for drinking is basically unacceptable in Czechia, it is normal in countries with deserts [17,19]. Even though people in Czechia have easy access to high-quality drinking water, they still believe that conserving water is important.

Another barrier towards using purified water in households is the lack of understanding of the water purification technology which further introduces fear from using purified water. People mostly feared residual pathogens, smell, colour, etc. However, a good water purification technology leaves almost none of such residuals. The "eww" effect of purified water is strong within the Czech public. If hygiene plays such a critical role in the acceptance of purified water, then people need to know that purified water is hygienically safe. The main target should be to persuade people to use purified water to flush toilets, which is hygienically safe. The government needs to explain this to the people to increase the acceptance for water purifying technologies. While technologically, water purification is a complex process, it can be simplified for the public and households [23,25].

We found that most household owners want to conserve water, but they do not know how to do it effectively. They had little knowledge on how to save more water and how they can personally contribute. Especially the acceptance of water-conserving technology that is easy and comfortable to use was over 80–90%. The government should build on that and support water-saving technologies. If that happens, we know that people will buy and use them. This is similar to other states where people also want to act pro-environmentally but do not know to implement it [20–27]. This calls for an information campaign that would educate the public and households on effective water conserving methods. If the government tells people how to specifically conserve water, then people will start acting on it. While similar campaigns already exist, they need to be locally targeted for a specific socio-geographic region [22].

Another discovery was that people who are willing to conserve and use purified water were not that different from those who are not willing to do so. They did not differ much in socio-economic status, age, gender, education, etc. This shows that both segments are a mix of different subgroups of people. Both groups were very heterogeneous, which makes it harder to target the specific group of people who are not yet conserving water but want to do so. This aligns well with other studies that analysed pro-environmental communication strategies. They found that targeting pro-environmentalists is difficult because they cannot be easily distinguished from the general public [3,10,13].

A key ingredient in targeting a group is trust in persons and public figures. We found that household owners mostly trusted people and figures close to them. This is well described in other studies as well—people trust someone they know personally [32]. This suggests that pro-environmental campaigns should be carried out by local majors. In addition, household owners also claimed to trust scientists and water management authorities. Other psychological studies have also confirmed it; however, trust in scientists is burdened with the error of social acceptability. People say that they trust scientists a lot, but in real life, they seldom follow their suggestions [32]. Conversely, people did not trust celebrities. However, celebrities are often used as speakers in communication campaigns. Psychological studies found that people do not want to admit that they are influenced by celebrities, while in fact, celebrities have large effect on opinions of the public [33].

This study has brought some interesting insights, but it has its limitations as well. The data we collected for this study are relatively new (1.5 years old); however, similar psychological and sociological studies age relatively quickly and could be potentially considered too old especially since in the last two years we have lived through major historical events like the COVID-19 pandemic, war in Ukraine, and financial inflation of 17% (in Czechia). Such events are known to change the perceptions of people [34].

We recommend six messages for the government to persuade household owners to conserve water and use purified water. These messages are mostly connected to the local environmental and climatic conditions, so anyone who wants to use them should adjust them to fit their geographical region. Whoever uses them should account for aspects like water scarcity, acceptance of water purification technologies, availability of drinking water, etc. These factors significantly affect the acceptance of water conservation actions and they differ among regions.

## 5. Conclusions

We conclude that household owners in Czechia are willing to conserve water, but they do not yet know how to do it effectively. They are less willing to use purified water mostly because of hygienic reasons.

*Recommendations for Communication with the Target Group*

Based on the motivations and barriers of household owners towards conserving water and using purified water in their households, we created the following six recommendations that the government should use to persuade homeowners to start conserving water and using purified water:

1. Water quality is perhaps an even more important problem than water quantity. Even though the industry and agriculture consume more water than households, the households can still contribute to water conservation. Every single person in a household can conserve water.
2. Households can effectively conserve water by using water-saving programs. In addition, economical programs for washing dishes or laundry are not difficult to operate. A large amount of water can be realistically saved, especially by flushing the toilet with a smaller volume of water or by saving water when brushing your teeth. Another possibility to save water is showering or dry cleaning instead of washing with water. By saving water, the household can also save money.
3. Use rainwater whenever possible. Due to its composition and temperature, rainwater is suitable for watering lawns and flower beds, just like tap or bottled water. Moreover, its use is not complicated, and an interesting amount of tap water can be saved during the year.
4. Water that undergoes treatment in the home using purification technologies is not significantly more harmful than ordinary tap water. It does not contain pathogens or remnants of the original impurities, is not harmful to health, does not smell, and does not have an unpleasant colour. If this purified water, coming from the sink or

household appliances, is used as utility water to flush the toilet, there is no risk of any significant health or other complications.

5. Installing water purification technologies is an investment that pays off in the long term, especially when building a new house or renovating an existing one. The financial return on this investment is around 10–15 years. In addition, by saving water, you will help protect water resources as a key strategic raw material, the value of which will rather increase in the future with the progress of climate change. On the other hand, it should be noted that water recycling technology is a financially demanding investment.

6. Filling swimming pools or garden ponds, and frequent watering of lawns, consumes a large amount of water. This is a problem especially in the summer when there is a significant local shortage of high-quality drinking water in many places. Swimming pools, ponds, and lawns have high water evaporation in summer temperatures, which further increases the already high water consumption for filling and watering. Households can conserve water by not using swimming pools and garden ponds in areas with drought.

**Author Contributions:** Conceptualization, methodology, validation, formal analysis, investigation, data curation, writing—original draft preparation, writing—review and editing, visualization, supervision, project administration, R.L. and J.R. All authors have read and agreed to the published version of the manuscript.

**Funding:** This research was funded by Operation Programme Research, Development and Education, European Structural and Investment Funds, and by the Ministry of Education, Youth and Sports of the Czech Republic, grant number/registration number CZ.02.2.69/0.0/0.0/18_054/0014660 as a part of the project "Setting the conditions and the environment for international and cross-sector cooperation".

**Institutional Review Board Statement:** The study was conducted in accordance with the Declaration of Helsinki of 1975 (https://www.wma.net/what-we-do/medical-ethics/declaration-of-helsinki/, accessed on 1 November 2022) and follows the Ethical codex (https://www.aapor.org/Standards-Ethics/AAPOR-Code-of-Ethics.aspx, accessed on 1 November 2022). The INESAN ethic review board does not require an additional ethics approval for such studies according to the ethical codex (https://inesan.eu/wp-content/uploads/2020/12/A_Eticky-kodex.pdf, accessed on 1 November 2022) because the institute holds a HRS4R HR Excellence in Research award (https://inesan.eu/en/hrs4r-2/, accessed on 1 November 2022). This award grants the highest level of ethical work carried by the researchers at this institute (https://www.euraxess.cz/jobs/hrs4r, accessed on 1 November 2022).

**Informed Consent Statement:** This study involved data collected from anonymous interviewees. All subjects gave their informed consent for inclusion before they participated in the study.

**Data Availability Statement:** The data used to support the findings of this study will be available from the corresponding author upon reasonable request. Since the data are owned by a third party, consent will be needed from this party as well.

**Acknowledgments:** We thank all interviewers for helping with the interviews.

**Conflicts of Interest:** The authors declare no conflict of interest. The funders had no role in the design of the study; in the collection, analyses, or interpretation of data; in the writing of the manuscript; or in the decision to publish the results.

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
