# Peer review of "Motivations of Households towards Conserving Water and Using Purified Water in Czechia"

_sustainability, doi:10.3390/su15032202_

Round 1

Reviewer 1 Report

It seems that the paper in its current form is weak and needs to be strengthened and rewritten. Addressing the below comments help improving this manuscript:   

Line 41. For the first time, the full term and its abbreviation should be written.

Line 49- It should be explained why you have focused on psychological and sociological factors.

The introduction has not explained the problem investigated in this study well. Therefore, the author should explain more information about the problem that this manuscript seeks to reduce.

The two sentences that follow are about the number of respondents and do not match each other. Be sure to explain sufficient explanations in this regard.

Line 72- Overall, 3 462 Czech residents were asked for an interview, and 1857 responded positively (54 %), and 1 856 were interviewed (others declined to participate).

Line 91- The data were collected in the Czech Republic in April-May 2021 using questionnaire surveys (1 800 respondents).

The part “2.2 Data collection tool (questionnaire)It should be rewritten very carefully.

What exactly has TPB been used for? It seems that the variables measured by the authors are not related to the TPB.

Please rewrite the spectrum (I mind – I do not mind, I agree – I disagree, a lot – not at all)

It is better to write complete discussions first and then explain practical and operational implications as much as possible.

Author Response

Reviewer 1

It seems that the paper in its current form is weak and needs to be strengthened and rewritten. Addressing the below comments help improving this manuscript:   

Line 41. For the first time, the full term and its abbreviation should be written.

Authors: Abbreviation was added (line 55)

Line 49- It should be explained why you have focused on psychological and sociological factors. 

The introduction has not explained the problem investigated in this study well. Therefore, the author should explain more information about the problem that this manuscript seeks to reduce.

Authors: The reason why we focused on psychological and sociological factors was added. The problem was also explained (lines 65-68).

The two sentences that follow are about the number of respondents and do not match each other. Be sure to explain sufficient explanations in this regard.

Line 72- Overall, 3 462 Czech residents were asked for an interview, and 1857 responded positively (54 %), and 1 856 were interviewed (others declined to participate).

Line 91- The data were collected in the Czech Republic in April-May 2021 using questionnaire surveys (1 800 respondents).

Authors: The number of respondents is 1 824, which was corrected in the whole manuscript (lines 11, 103, 116, 124, 184, 205, 220, 237, 244, 256, 265).

The part “2.2 Data collection tool (questionnaire)” It should be rewritten very carefully.

What exactly has TPB been used for? It seems that the variables measured by the authors are not related to the TPB.

Authors: The methods have been rewritten (lines 89-109, 116-121, 128-159, 162-169)

Please rewrite the spectrum (I mind – I do not mind, I agree – I disagree, a lot – not at all)

Authors: The methods have been rewritten (lines 89-109, 116-121, 128-159, 162-169)

It is better to write complete discussions first and then explain practical and operational implications as much as possible.

Authors: We moved the practical implications right after the discussion (lines 350-388).

Reviewer 2 Report

The manuscript entitled “Motivation of households towards conserving water and using purified water in the Czech Republic” gives an insight into the people’s perception of conserving water and using purified water in the current scenario. The methods used are satisfactory however, the manuscript is found with errors that need to be corrected, both in terms of technical writing and presentation. For now, the manuscript resembles a report. Authors are suggested to have a look at the highlighted texts in the attached pdf. Additionally, I would like to add some of the suggestions before the publication. Please consider this before the second review process.

1.     Authors are suggested to define what is conserved water and purified water whether its recycled water, filtered water, or water treated with RO in terms of the Czech Republic. The definition would justify the statement in lines 35-38.

2.     Please define what is PR in Line 41.

3.     The authors also suggested adding the source of water being used in the area and their conditions in the method and linking this into the discussion section.

4.     Please define 4 points and 5 points.

5.     Please move the recommendation at the end of the manuscript to right after the result. Make recommendations to the point and effective for the policymakers.

6.     Discussion needs to be modified as with such enormous results the discussion can be made more interesting to the readers.

7.     Figure 1. Please add a scale and label to the figure.

8.     Figure 2. Please define figure a and figure b in the caption. The title of the figures is the same for both making it a bit difficult to visualize. Also please change the legend making it specific to each figure.

Author Response

Reviewer 2

The manuscript entitled “Motivation of households towards conserving water and using purified water in the Czech Republic” gives an insight into the people’s perception of conserving water and using purified water in the current scenario. The methods used are satisfactory however, the manuscript is found with errors that need to be corrected, both in terms of technical writing and presentation. For now, the manuscript resembles a report. Authors are suggested to have a look at the highlighted texts in the attached pdf. Additionally, I would like to add some of the suggestions before the publication. Please consider this before the second review process.

  1. Authors are suggested to define what is conserved water and purified water whether its recycled water, filtered water, or water treated with RO in terms of the Czech Republic. The definition would justify the statement in lines 35-38.
  2. Please define what is PR in Line 41.
  3. The authors also suggested adding the source of water being used in the area and their conditions in the method and linking this into the discussion section.
  4. Please define 4 points and 5 points.
  5. Please move the recommendation at the end of the manuscript to right after the result. Make recommendations to the point and effective for the policymakers.
  6. Discussion needs to be modified as with such enormous results the discussion can be made more interesting to the readers.
  7. Figure 1. Please add a scale and label to the figure.
  8. Figure 2. Please define figure a and figure b in the caption. The title of the figures is the same for both making it a bit difficult to visualize. Also please change the legend making it specific to each figure.

Authors:

  1. The explanation for conserving water and using purified water was added (lines 31-33).
  2. The definition of PR was added (line 55).
  3. The source of water being used in the area was added (lines 84-88). It was also linked to the discussion section (lines 289-291).
  4. The methods have been rewritten (lines 89-109, 116-121, 128-159, 162-169)
  5. The recommendations were moved to the end of the manuscript (lines 350-388). They were made to the point and effective for the policymakers. However, they need to stay generalized to some point, as they need to be useful globally in different geographical environments.
  6. The discussion section was widened (lines 275-280, 284-286, 394-300, 305-308, 310-312, 315-317).
  7. Scale and label were added to the figure (Figure 1).
  8. We defined figures A and B in the figure caption. We also changed the titles and legends for both figures (Figure 2).

Reviewer 3 Report

This is an interesting study on water usage and willingness to adopt more sustainable behaviors toward water conservation. The results are interesting although they do not advance much in the current knowledge on the specific subject under study. The study has also some serious flaws that compromise the chance of a direct positive evaluation.

Although the questionnaire and techniques are sound, the authors claim specific methods for data analysis (factor analysis) that are not clear in the manuscript. The discussion can also be explored in much more detail, considering this is a topic already well supported by social science studies.

Overall, I would not reject the manuscript due to the local relevance of the results but recommend a major revision.

Below, I outline some detailed comments that may, hopefully, help the authors improve the study.

Introduction section:

Page 1, line 27: This sentence needs a reference.

Page 2, line 49: Although I understand where the authors are pointing, I don’t know if ‘psychologically sociological study’ is a correct statement. Please make sure the it exists and it is correct.

Not clear who the target group is (aside from being a sample of the Czech Republic). Should be clearly stated from the start. A subheading referring to the aim / objective / questions of the study should help to clearly outline the purpose of the study and help guide the reader throughout 

The authors need also to further explore the behaviors of saving water and using purified water. The former is straightforward but still needs more detail. The second, when living in a progressive and XXI century European country, needs clarification. I understand the authors’ intention but getting an extra source of water that does not depend on depleting water reservoirs is a different rooted behavior from that of, e.g., collecting rain water for watering plants. This is even more evident when the sampled universe does not face serious drought threat as other European countries.

Methods section:

Page 5, line 96: Not clear how the TPB is here associated. The authors should address this aspect in more detail.

Page 5, line 111: the authors state ‘reasons not to conserve water’ but shouldn’t it be ‘reasons to conserve water’, once the examples are scarcity-wise?

With such a long questionnaire, and for the sake of replication, it is important to state the average time spent to answer it.

Data analysis:

Not clear how factor analysis was used. The authors need to explore this in further detail, as well as show the results of the factor analysis. It seems the analysis is merely based on descriptive data.  

Results section:

Page 6, lines 139-141: these two sentences are not results. They should be present in the discussion section.

Figure captions: the authors refer to 1800 respondents but the sample was 1856. If it is a rounded number, it should be stated as so. If not, the authors should explain in text the decrease in the number of respondents.

Page 6, lines 149-155: the sentence ‘The groups of people who are interested and not interested in conserving water and using purified water do not significantly differ in gender, age, socio-economic status, occupation (type of job), or type of living.’, if framed like this, lacks statistical information on the statements.

Page 9, line 185: I think the authors want to say ‘not hygienic-related’ and not the double negative ‘not unhygienic’

Page 11, line 209: the authors refer to car users when it should read home owners. Maybe adapted from a different study?

Page 12, line 216: again, car users…

Page 12, line 220: the caption also refers to car users

Pages 12-13, lines 223-260: this entire subheading should not be present in the results section but in the discussion section.

Discussion section:

Page 13, line 282: ‘about saves more water’ should probably be ‘about saving more water’

Page 13, line 294: ‘key ingrediency’ should read ‘key ingredient’

Page 13, line 301: The English needs to be improved. It is not celebrities that had little trust. It is the people that does not trust celebrities.

Page 13, lines 301-304: the findings and the rational are contradictory.

Author Response

Reviewer 3

This is an interesting study on water usage and willingness to adopt more sustainable behaviors toward water conservation. The results are interesting although they do not advance much in the current knowledge on the specific subject under study. The study has also some serious flaws that compromise the chance of a direct positive evaluation.

Although the questionnaire and techniques are sound, the authors claim specific methods for data analysis (factor analysis) that are not clear in the manuscript. The discussion can also be explored in much more detail, considering this is a topic already well supported by social science studies.

Overall, I would not reject the manuscript due to the local relevance of the results but recommend a major revision.

Below, I outline some detailed comments that may, hopefully, help the authors improve the study.

Authors: This study brings new data from the Czech Republic where similar study does not yet exist. This should make it interesting for international audience. We removed the flaws in the study. The methods have been rewritten (lines 89-109, 116-121, 128-159, 162-169). The discussion section was widened (lines 275-280, 284-286, 394-300, 305-308, 310-312, 315-317).

Introduction section:

Page 1, line 27: This sentence needs a reference.

Authors: Reference was added (line 27)

Page 2, line 49: Although I understand where the authors are pointing, I don’t know if ‘psychologically sociological study’ is a correct statement. Please make sure the it exists and it is correct.

Authors: The sentence was corrected to sociological study (line 64)

Not clear who the target group is (aside from being a sample of the Czech Republic). Should be clearly stated from the start. A subheading referring to the aim / objective / questions of the study should help to clearly outline the purpose of the study and help guide the reader throughout 

Authors: The subheading was added and the target group was defined in the introduction (lines 71-80).

The authors need also to further explore the behaviors of saving water and using purified water. The former is straightforward but still needs more detail. The second, when living in a progressive and XXI century European country, needs clarification. I understand the authors’ intention but getting an extra source of water that does not depend on depleting water reservoirs is a different rooted behavior from that of, e.g., collecting rain water for watering plants. This is even more evident when the sampled universe does not face serious drought threat as other European countries.

Authors: The introduction on water saving and usage of purified water was expanded (lines 37-43, 45-50, 52-53)

Methods section:

Page 5, line 96: Not clear how the TPB is here associated. The authors should address this aspect in more detail.

Authors: The methods have been rewritten (lines 89-109, 116-121, 128-159, 162-169)

Page 5, line 111: the authors state ‘reasons not to conserve water’ but shouldn’t it be ‘reasons to conserve water’, once the examples are scarcity-wise?

Authors: Yes, that is true, the sentence was corrected (lines 140-141)

With such a long questionnaire, and for the sake of replication, it is important to state the average time spent to answer it.

Authors: The time needed to complete the questionnaire was added (lines 107-108).

Data analysis:

Not clear how factor analysis was used. The authors need to explore this in further detail, as well as show the results of the factor analysis. It seems the analysis is merely based on descriptive data.  

Authors: We added the results of the factor analysis to the results section (lines 213-215, 229-232, 251-251)

Results section:

Page 6, lines 139-141: these two sentences are not results. They should be present in the discussion section.

Authors: Both sentences were removed from the results section.

Figure captions: the authors refer to 1800 respondents but the sample was 1856. If it is a rounded number, it should be stated as so. If not, the authors should explain in text the decrease in the number of respondents.

Authors: The correct number of respondents (1 824) was added to the text (lines 11, 103, 116, 124, 184, 205, 220, 237, 244, 256, 265).

Page 6, lines 149-155: the sentence ‘The groups of people who are interested and not interested in conserving water and using purified water do not significantly differ in gender, age, socio-economic status, occupation (type of job), or type of living.’, if framed like this, lacks statistical information on the statements.

Authors: The information on statistical insignificance was added (lines 190-191).

Page 9, line 185: I think the authors want to say ‘not hygienic-related’ and not the double negative ‘not unhygienic’

Authors: The sentence was corrected (line 224). 

Page 11, line 209: the authors refer to car users when it should read home owners. Maybe adapted from a different study?

Authors: The sentence was corrected (line 259)

Page 12, line 216: again, car users…

Authors: The sentence was corrected (line 265)

Page 12, line 220: the caption also refers to car users

Authors: The sentence was corrected (line 250)

Pages 12-13, lines 223-260: this entire subheading should not be present in the results section but in the discussion section.

Authors: The section was moved to conclusions (lines 350-388).

Discussion section:

Page 13, line 282: ‘about saves more water’ should probably be ‘about saving more water’

Authors: The sentence was corrected (line 304)

Page 13, line 294: ‘key ingrediency’ should read ‘key ingredient’

Authors: The sentence was corrected (line 322)

Page 13, line 301: The English needs to be improved. It is not celebrities that had little trust. It is the people that does not trust celebrities.

Authors: The sentence was corrected (line 329).

Page 13, lines 301-304: the findings and the rational are contradictory.

Authors: The sentence was rewritten (line 332).

Round 2

Reviewer 1 Report

The aim of the paper is analysis of perceptions of the public in Czechia towards water conservation and purified water. It is acceptable for publication after the editing of the following minor comments.

Line 31-33 need references.

The map should be drawn with more quality and details.

Line 204 "The data were collected in Czechia in April-May 2021 using questionnaire surveys" have to be in the methodology.

Author Response

The aim of the paper is analysis of perceptions of the public in Czechia towards water conservation and purified water. It is acceptable for publication after the editing of the following minor comments.

Line 31-33 need references.

Authors: the references were added (lines 31-33).

The map should be drawn with more quality and details.

Authors: we added a higher quality map (Figure 1)

Line 204 "The data were collected in Czechia in April-May 2021 using questionnaire surveys" have to be in the methodology.

Authors: the sentence is there (rephrased, lines 89-90).

Reviewer 3 Report

Overall, this version of the manuscript has improved considerably.

There is, nevertheless, the factor analysis topic that, imho, needs to be clearly described. The authors, for the sake of replication, need to briefly explain how the exploratory factor analysis was performed.

Author Response

Overall, this version of the manuscript has improved considerably.

There is, nevertheless, the factor analysis topic that, imho, needs to be clearly described. The authors, for the sake of replication, need to briefly explain how the exploratory factor analysis was performed.

Authors: The description of the factor analysis was added (lines 167-174).